# Geographical Variation Shapes Nutritional Metabolite Profile and Food Functionality of *Houttuynia cordata*

**DOI:** 10.3390/metabo15110701

**Published:** 2025-10-29

**Authors:** Yuanyuan Zhang, Xuelang Fu, Jinqun Zhu, Yu Gui, Huilin Huang, Yangye Liao, Yanping Mao, Hui Tian, Lei Liu

**Affiliations:** 1College of Biology and Pharmacy, Mianyang Normal University, Mianyang 621000, China; 199774@mtc.edu.cn (Y.Z.); sya0417@stu.mtc.edu.cn (J.Z.); 2414162912@stu.mtc.edu.cn (Y.G.); yangyeliao0808@stu.mtc.edu.cn (Y.L.); 2School of Science Bioinformatics, Xi’an Jiaotong-Liverpool University, Suzhou 215123, China; xuelang.fu22@student.xjtlu.edu.cn; 3College of Life Science, Mianyang Normal University, Mianyang 621000, China; huanghuiling123@stu.mtc.edu.cn (H.H.); yanpingmao@mtc.edu.cn (Y.M.)

**Keywords:** *Houttuynia cordata*, geographical variation, cultivation optimization, network pharmacology, functional foods

## Abstract

**Background/Objectives:** *Houttuynia cordata* Thunb., a widely consumed vegetable and traditional food in Asia, possesses significant nutritional value. However, the impact of geographical origin on its nutritional metabolite composition, crucial for food quality and functionality, remains unclear. **Methods:** Here, we conducted a comprehensive metabolomic analysis of *H. cordata* cultivated across six major Chinese regions (Yunnan, Guangxi, Guizhou, Sichuan, Chongqing, Hubei) using UPLC-MS/MS. **Results and Conclusions:** We identified 496 nutritional metabolites, predominantly amino acids and derivatives (53.23%). Key bioactive carbohydrates, maltotriose and maltitol, exhibited distinct geographical accumulation patterns: maltotriose was significantly enriched in Yunnan (YN), while maltitol accumulated predominantly in Sichuan (SC). Integrated transcriptomic analysis linked this variation to the differential expression of starch metabolism genes (GBE1/glgB, α-amylases, β-amylases). Bioinformatic evaluation suggested potential health-related functionalities associated with these metabolites. These findings provide critical insights into the geographical determinants of *H. cordata* nutritional quality and functional properties. They offer a scientific foundation for optimizing cultivation practices based on regional advantages and developing *H. cordata* as a region-specific, high-value vegetable and functional food ingredient targeting distinct nutraceutical applications.

## 1. Introduction

*Houttuynia cordata*, a perennial herb of the *Saururaceae* family, is an important edible and medicinal species, particularly popular as a vegetable in southwestern China [1,2]. Its tender stems and leaves are consumed fresh or prepared in various dishes, and it is also processed into beverages, teas, and preserved foods, meeting growing consumer demand for health-oriented products [3,4]. Beyond its culinary uses, *H. cordata* possesses remarkable therapeutic properties, traditionally employed to treat pulmonary abscess, dysentery, urinary infections, and inflammation [5,6]. Modern studies have identified bioactive constituents such as volatile oils, flavonoids, alkaloids, and polysaccharides, which contribute to its antimicrobial, antiviral, anti-inflammatory, antioxidant, and immunomodulatory effects [2,7,8,9,10,11,12]. These properties support its application in managing respiratory, urinary, and digestive infections, as well as in immune-enhancing nutraceuticals [13].

Given its dual significance, comprehensive research on *H. cordata* is of substantial practical importance. Environmental factors, such as climate, soil, and light vary significantly across geographical regions and critically influence plant growth, metabolite biosynthesis, and product quality [14]. Increasing market demand and expanded cultivation, exemplified by regional specialty production in Guizhou [15], highlight the need to understand how geographical origin affects its chemical composition. This is essential for quality standardization, cultivation optimization, and functional application [1,13,16,17].

*H. cordata* is cultivated across six Chinese regions, Yunnan, Guangxi, Guizhou, Sichuan, Chongqing, and Hubei, each with distinct climatic conditions that create varying growth suitability [18]. Sichuan, Chongqing, and Guizhou represent high-suitability zones due to favorable temperature and moisture regimes [15,16,18,19], while Hubei and Guangxi are medium- to low-suitability areas [20,21]. Yunnan, though of lowest general suitability, may promote unique bioactive compound synthesis under environmental stress [22].

Plant metabolites serve as fundamental building blocks for plant survival and as key sources of human nutrients and pharmaceuticals [23,24]. Amino acids function as nitrogen reserves and flavor contributors, with compounds like GABA and rosmarinic acid offering neuroactive and anti-allergic benefits [25,26]. Nucleotides act as energy carriers and immunomodulatory precursors, as seen in bamboo shoots and shiitake mushrooms [27]. Carbohydrates, such as inulin and konjac glucomannan, provide prebiotic and metabolic health benefits [28]. Vitamins, like the high vitamin C in bell peppers, form potent antioxidant systems [29], while steroidal compounds, including ginsenosides and andrographolide, exhibit anti-inflammatory and antiviral activities [30,31,32].

This study aims to comprehensively characterize the nutritional metabolite profile of *Houttuynia cordata* cultivated across six major geographical regions (Yunnan, Guangxi, Guizhou, Sichuan, Chongqing, and Hubei) using UPLC-MS/MS. By integrating transcriptomic sequencing and network pharmacology, we further seek to elucidate the geographical influences on metabolite accumulation, explore the underlying biosynthetic and regulatory mechanisms, and evaluate the potential health-related functionalities. This multi-omics investigation provides a scientific foundation for optimizing cultivation strategies and advancing the development of *H. cordata* as a high-value, region-specific functional food.

## 2. Materials and Methods

### 2.1. Plant Materials

To precisely evaluate the effects of geographical variation on nutritional metabolites while minimizing genotypic influences, a uniform transplanting experiment was conducted. A single genotype of *H. cordata* (accession No. 7) was utilized as the experimental material. All plant materials were initially propagated and pre-cultured under standardized conditions at Mianyang Normal University (Mianyang City, Sichuan Province) to ensure uniform initial growth status. On 1 July 2023, healthy and uniformly grown individuals of *H. cordata* accession No. 7 were transplanted to six geographically distinct regions across China: Sichuan (SC) (Mianyang Normal University, Mianyang City), Chongqing (CQ) (Chongqing University, Chongqing City), Guangxi (GX) (Guangxi University, Nanning City), Guizhou (GZ) (Guizhou University, Guiyang City), Hubei (HB) (Huazhong Agricultural University, Wuhan City), and Yunnan (YN) (Honghe University, Mengzi City). Across all locations, plants were grown under consistent cultivation protocols using standardized plastic pots (28 cm diameter × 18.5 cm depth) filled with a uniform 1:1 (*v*/*v*) peat soil-garden soil substrate mixture. After one full year of growth under local environmental conditions (on 30 June 2024), the aboveground parts (stems and leaves) of the plants were harvested. The harvested samples were immediately frozen in liquid nitrogen and subsequently stored at −80 °C for further biochemical analyses.

### 2.2. Analysis of LC-MS/MS Platform

#### 2.2.1. Sample Preparation and Extraction

The biological samples were subjected to vacuum freeze-drying using a lyophilizer Scientz-100F (SCIENTZ, Ningbo, China). Subsequently, the freeze-dried samples were homogenized into powder form (30 Hz, 1.5 min) with a grinder MM 400, (Retsch, Haan, Germany). Following this, 50 mg of powdered sample was precisely weighed using an electronic balance MS105DU (Mettler-Toledo Group, Shanghai, China) and mixed with 1200 μL of pre-cooled (−20 °C) 70% methanolic aqueous solution containing internal standards. The mixture was vortexed for 30 s at 30 min intervals (6 cycles total). After centrifugation at 14,000× *g* for 3 min, the supernatant was carefully aspirated, filtered through a 0.22 μm microporous membrane, and transferred to injection vials for subsequent UPLC-MS/MS analysis.

#### 2.2.2. UPLC Conditions

The sample extracts were analyzed using a UPLC-ESI-MS/MS system (UPLC: ExionLC^TM^ AD, https://sciex.com.cn/ (accessed on 20 July 2024); MS: Applied Biosystems 6500 QTRAP, https://sciex.com.cn/ (accessed on 22 July 2024)). The analytical conditions were as follows: An Agilent SB-C18 column (1.8 µm, 2.1 × 100 mm) was used with a mobile phase consisting of solvent A (0.1% formic acid in ultrapure water) and solvent B (0.1% formic acid in acetonitrile). The gradient elution program started with 95% A and 5% B, followed by a linear gradient to 5% A and 95% B over 9 min. This ratio was maintained for 1 min, after which the composition returned to 95% A and 5% B within 1.1 min and was equilibrated for 2.9 min. The analysis was performed with a flow rate of 0.35 mL/min, a column temperature of 40 °C, and an injection volume of 2 μL. The column effluent was directly introduced into the ESI-triple quadrupole-linear ion trap (QTRAP)-MS for detection.

#### 2.2.3. ESI-Q TRAP-MS/MS

The ESI source parameters were configured as follows: source temperature 500 °C; ion spray voltage (IS) ± 5500 V (positive/negative ion mode); gas pressures for ion source gas I (GSI), gas II (GSII), and curtain gas (CUR) at 50, 60, and 25 psi, respectively; collision-activated dissociation (CAD) set to high. QQQ scans were performed in MRM (multiple reaction monitoring) mode with nitrogen collision gas at medium pressure. Declustering potential (DP) and collision energy (CE) for individual MRM transitions were optimized through iterative parameter adjustments. During analysis, specific MRM transitions were dynamically monitored for each chromatographic period based on metabolite elution profiles.

#### 2.2.4. Selection of Differential Metabolites with KEGG Annotation and Enrichment Analysis

Differential metabolites were identified based on a variable importance in projection (VIP) score > 1 and an absolute log2-fold change (|log2FC|) ≥ 1.0. The annotated metabolites were first mapped to the KEGG COMPOUND database (https://www.kegg.jp/kegg/compound/ (accessed on 25 October 2024)) for functional classification. Subsequently, these metabolites were systematically aligned with the KEGG PATHWAY database (https://www.kegg.jp/kegg/pathway.html (accessed on 27 October 2024)) to perform pathway enrichment analysis.

### 2.3. RNA-Sequencing

#### 2.3.1. RNA Extraction and Detection

Plant RNA was extracted using ethanol precipitation coupled with CTAB-PBIOZOL methodology. Following extraction, the RNA was solubilized in 50 µL of DEPC-treated water. Total RNA was then quantified using a Qubit Fluorometer (Thermo Fisher Scientific, Waltham, MA, USA) and its integrity verified through capillary electrophoresis analysis on a Qsep400 bioanalyzer (BiOptic Inc., Taiwan, China).

#### 2.3.2. mRNA Library Construction

The library preparation protocol proceeded as follows: Polyadenylated mRNAs were first enriched using oligo (dT) magnetic beads, leveraging the polyA tail structure of eukaryotic transcripts. Subsequently, purified mRNAs were fragmented under optimized thermal conditions using a fragmentation buffer. First-strand cDNA synthesis was then performed with random hexamer primers, followed by second-strand cDNA generation employing a strand-specific strategy via dUTP substitution (replacing dTTPs) coupled with end repair and dA-tailing. Adaptor ligation was executed prior to size selection (250–350 bp inserts) using magnetic bead purification. PCR amplification of ligated products was subsequently conducted, followed by final purification in nuclease-free water. The resulting libraries underwent rigorous quality control, with concentration quantified via a Qubit Fluorometer and fragment size distribution validated using a Qsep400 bioanalyzer.

#### 2.3.3. Sequencing Run

Following successful library qualification, pooled libraries were sequenced on an Illumina platform using 150 bp paired-end reads, with pooling ratios determined by both effective library concentrations and required sequencing depth. The sequencing-by-synthesis approach employed fluorescently labeled dNTPs (four types), DNA polymerase, and immobilized primers within flow cell channels. During cluster amplification, each incorporated dNTP released characteristic fluorescence signals, which were optically detected and subsequently converted to base calls through integrated software, thereby determining the nucleotide sequence of each DNA fragment.

#### 2.3.4. Gene Function Annotation and Differential Analysis

Transcript sequences were functionally annotated through sequential alignment against seven major databases: KEGG, NR, Swiss-Prot, GO, COG/KOG, and TrEMBL using DIAMOND, while Pfam domain analysis was performed on amino acid sequences via HMMER (v3.3.2). For differential expression analysis, DESeq2 was employed to identify statistically significant genes, with adjusted *p*-values derived from the Benjamini–Hochberg false discovery rate (FDR) correction. Genes meeting the dual thresholds of FDR < 0.05 and absolute log2-fold change (|log2FC|) ≥ 1 were classified as differentially expressed.

### 2.4. Pharmacological Efficiency Analysis of Metabolites

#### 2.4.1. Prediction of Targets

The SMILES (Simplified Molecular Input Line Entry System) files of metabolites identified in *H. cordata* from diverse geographical regions through metabolomic analysis were retrieved from the PubChem database (https://pubchem.ncbi.nlm.nih.gov/ (accessed on 5 December 2024)). Putative metabolite targets were predicted via the Swiss Target Prediction database (http://www.swisstargetprediction.ch/ (accessed on 10 December 2024)) with species restricted to Homo sapiens and a probability threshold >0.1. Finally, a metabolite-target interaction network was generated and visualized using Cytoscape 3.9.1 software.

#### 2.4.2. Analysis of Protein–Protein Interaction Network

Protein–protein interaction (PPI) analysis of putative targets was conducted using the STRING 12.0 database (https://cn.string-db.org/ (accessed on 22 December 2024)) with species limited to Homo sapiens and a minimum interaction score > 0.9 (high confidence). Overlapping targets were further analyzed through the cytoHubba plugin in Cytoscape 3.9.1, where core candidate targets were identified by selecting nodes with Maximal Clique Centrality (MCC) scores exceeding their mean values. Finally, a PPI network of the core targets was constructed using Cytoscape 3.9.1 software.

#### 2.4.3. GO and KEGG Enrichment Analysis

The core candidate targets identified through the analysis underwent Gene Ontology (GO) enrichment analysis (biological processes, cellular components, and molecular functions) and Kyoto Encyclopedia of Genes and Genomes (KEGG) pathway analysis using the STRING 12.0 database (https://cn.string-db.org/ (accessed on 15 January 2025)). These enrichment results were then visualized through R version 4.2.0 following standard analytical workflows.

#### 2.4.4. Network of Efficacious Metabolites–Target–Pathway–Disease Interactions

Metabolites and their disease associations were sourced from the Traditional Chinese Medicine Systems Pharmacology Database (TCMSP, https://www.tcmsp-e.com/tcmsp.php (accessed on 15 January 2025)) and PubChem database (https://pubchem.ncbi.nlm.nih.gov/ (accessed on 15 January 2025)). Disease-associated targets of efficacious metabolites were subsequently identified from the GeneCards database (https://www.genecards.org/ (accessed on 16 January 2025)) with a relevance score threshold > 5. Ultimately, a comprehensive network diagram depicting interrelationships between efficacious metabolites, targets, pathways, and diseases was generated using Cytoscape 3.9.1 software.

### 2.5. Molecular Docking Simulation

The three-dimensional (3D) structures of the key target proteins were acquired from the UniProt database (https://www.uniprot.org/ (accessed on 6 February 2025)), whereas the candidate metabolites’ 3D structures were sourced from the PubChem database (https://pubchem.ncbi.nlm.nih.gov/ (accessed on 6 February 2025)). Molecular docking simulations were subsequently conducted through the CB-DOCK2 platform (https://cadd.labshare.cn/cb-dock2/index.php (accessed on 8 February 2025)) to assess binding interactions, with visualization performed using Discovery Studio 2019.

### 2.6. Data Statistics

Data analysis was performed using Office 2021 (Microsoft Corporation, Redmond, WA, USA). Visualization outputs including upset plots, pie charts, Principal Component Analysis (PCA), and petal Venn diagrams were generated through R version 3.5.1. Subsequently, K-means clustering analysis, circular heatmaps, and standard heatmaps were created using R version 4.2.0. Finally, bubble plot visualization was implemented with R version 4.2.2.

## 3. Results

### 3.1. General Overview of Metabolites

Based on *H. cordata* cultivated across six geographical regions, 496 nutritional metabolites were identified. Their relative abundance profiles were visualized through a heatmap (Figure 1A), which distinctly revealed region-specific accumulation patterns.

These metabolites were taxonomically classified into five categories (Figure 1B): amino acids and derivatives, carbohydrates, nucleotides and derivatives, steroids, and vitamins. Notably, amino acids and derivatives constituted the predominant category (264 metabolites, 53.23%), while steroids represented the smallest group (10 metabolites, 2.02%).

Quantitative analysis of metabolite distribution across regions (Figure 1C) demonstrated minimal variation in total metabolite counts. Chongqing (CQ) exhibited the highest count (484 metabolites), followed by Yunnan (YN), Hubei (HB), Guangxi (GX), and Guizhou (GZ). Sichuan (SC) showed the lowest count (474 metabolites). Cross-regional analysis identified 434 metabolites shared across all six regions, with the other 62 metabolites overlapping in 2–5 regions. Importantly, no region-specific metabolites were detected.

Principal component analysis (PCA) of the metabolite profiles (Figure 1D) yielded two components explaining 59.91% of total variance (PC1: 37.59%; PC2: 22.32%). Tight clustering of triplicate biological replicates confirmed methodological reproducibility. Geographically distinct clustering patterns emerged: YN and CQ formed separate clusters, while GZ, HB, SC, and GX exhibited spatial proximity, particularly with GZ and HB forming a distinct subgroup. These patterns collectively reflect region-dependent compositional characteristics of nutritional metabolites.

### 3.2. Global Profiling of Differential Metabolites

Using differential accumulation criteria, we identified 452 differentially accumulated metabolites (DAMs). These DAMs were visualized through a heatmap (Figure 2A), revealing distinct accumulation patterns across geographical regions. Notably, all 10 identified steroids qualified as DAMs, while the remaining four metabolite categories exhibited varying proportions of non-differential metabolites excluded during screening.

To further characterize differential patterns, K-means clustering analysis (Figure 2B) partitioned DAMs into 10 subclasses with distinct accumulation profiles. Subclass 7 contained the largest DAM cohort (132 metabolites), showing high accumulation in Yunnan (YN) versus low abundance in other regions. Subclasses 6 (42 DAMs) and 8 (28 DAMs) demonstrated similar YN-specific enrichment but with additional high accumulation in Chongqing (CQ) and Guangxi (GX), respectively. Subclass 5 ranked second in DAM quantity (65 metabolites), exhibiting CQ-specific accumulation with minimal presence elsewhere. Remaining subclasses contained smaller DAM cohorts (20–40 metabolites per group), each displaying unique geographical accumulation signatures.

### 3.3. Comparative Analysis of DAMs Across Different Comparison Groups

To systematically analyze geographical variation in *H. cordata* metabolites, we conducted comparative profiling of differentially accumulated metabolites (DAMs) across 15 comparison groups. Bubble plot visualization (Figure 3A) revealed distinct DAM accumulation patterns among groups. Substantial variation emerged in DAM quantities across comparisons (116–300 metabolites), with the five most divergent groups being SC_vs_YN (300 DAMs), YN_vs_CQ (271), YN_vs_HB (270), YN_vs_GZ (268), and YN_vs_GX (265). Notably, all top five comparisons involved Yunnan (YN), indicating pronounced metabolic divergence between YN-grown materials and other regions. Conversely, GZ_vs_HB exhibited minimal divergence (116 DAMs), consistent with PCA clustering patterns. Comparative analysis further demonstrated elevated DAM accumulation in YN and CQ relative to GX, GZ, HB, and SC.

Petal Venn diagram analysis (Figure 3B) identified no universally shared DAMs across all 15 comparisons. However, 11 comparison groups (excluding CQ_vs_HB, GZ_vs_HB, SC_vs_GZ, and YN_vs_GX) contained unique DAMs. Particularly, SC_vs_HB showed the highest count of unique DAMs (5 metabolites), highlighting region-specific metabolic signatures.

### 3.4. Analysis of Putative Targets for Metabolites

We analyzed putative targets of candidate differential metabolites uniquely identified through metabolomic screening, revealing 138 targets associated with 15 specific metabolites. A regulatory network mapping metabolites to their corresponding targets was constructed to visualize these relationships (Figure 4A). Subsequently, protein–protein interaction (PPI) network analysis identified 26 key candidate targets with high connectivity (Figure 4B), suggesting their central roles in mediating metabolic regulatory effects. The PPI topology further clarified functional hierarchies and interaction patterns among these prioritized targets.

### 3.5. Enrichment Analysis of Candidate Targets

The 26 core targets exhibited significant GO enrichment across three ontological categories. Cellular Component (CC) analysis identified 21 subcellular localizations, with apoptotic execution complexes (Caspase complex, Death-inducing signaling complex) co-occurring with neuronal plasticity structures (Dendritic spine, Postsynapse), suggesting dual pharmacological mechanisms: targeted modulation of apoptosome assembly for pathological cell clearance and neuroprotection through dynamic reorganization of dendritic architectures. Molecular Function (MF) enrichment revealed three enzymatic activity hubs: (1) cysteine-type endopeptidase cascades governing apoptosis execution, (2) metabolic regulatory enzymes involving glycogen phosphorylase and pyridoxal phosphate binding, and (3) signal transduction interfaces through phospholipase activation and receptor binding. This functional triad aligns with the polypharmacological mode of action defined by FDA guidelines for multi-target modulators. Biological Process (BP) analysis of the top 8 among 205 enriched pathways established a four-tier regulatory network: (1) cell fate determination via crosstalk between apoptosis and pyroptosis, (2) metabolic-immune interplay mediated by glycogen catabolismand IL-6 signaling bridges, (3) environmental stress responses via shared TLR4-NFκB pathways in fluid shear stress and LPS response, and (4) neuroendocrine coordination through PI3K-Akt-mediated convergence of TRK receptor and GnRH signaling. Three druggable functional modules emerged from integrated GO analysis (Figure 5A, Appendix A): Apoptosis-Metabolism Regulator: Synergistic control of caspase complexes, glycogenolysis, and phospholipase activation demonstrates therapeutic potential for diabetic complications through dual metabolic/apoptotic modulation. Neuro-Immune Switch: Coordinated dendritic spine remodeling, TLR signaling, and TRK receptor crosstalk suggests novel interventions for neurodegenerative disorders like Alzheimer’s disease. Mechano-Chemical Sensor: Concurrent enrichment in fluid shear stress responses, cytoskeletal reorganization, and ion channel regulation reveals actionable targets for hemodynamic interventions in atherosclerosis. Key target validation confirmed Caspase-3/7 as central apoptosis executors (CC-BP overlap), GPBB as a metabolic nexus (MF-BP linkage), and TRPV1 channels as dual mechanical/inflammatory transducers, highlighting their translational relevance.

KEGG pathway enrichment analysis of the 26 identified core targets revealed 66 significantly enriched pathways (*p* < 0.05). The comprehensive network demonstrated three-tier pharmacological correlations, including Anti-infective mechanisms: Pathogen-specific pathways (Salmonella infection, Legionellosis, Shigellosis) intersected with immune regulation pathways (NOD-like receptor signaling, C-type lectin receptor signaling), suggesting broad-spectrum antimicrobial potential through dual pathogen-targeting and host immunity modulation. Neoplastic intervention: Cancer-associated pathways (p53 signaling, Viral carcinogenesis, Platinum drug resistance) converged with cellular homeostasis pathways (Apoptosis, Necroptosis), indicating multi-target oncological effects through simultaneous induction of tumor cell death and drug resistance reversal. Metabolic regulation: Insulin signaling pathway showed nodal connections with both glucagon signaling and inflammatory pathways (IL-17 signaling, TNF signaling), proposing a unique therapeutic strategy for diabetes complications via metabolic-inflammatory crosstalk regulation. The top 20 enriched pathways were visualized through a circular enrichment plot (Figure 5B, Appendix A), revealing three pharmacological clusters: Modulates cellular mechanotransduction: Critical for cardiovascular drug efficacy against fluid shear stress-induced atherosclerosis. Targets metabolic memory in diabetic complications: Explains hepatoprotective effects through lipid signaling modulation. Reveals endocrine therapy optimization potential through growth factor receptor cross-talk regulation. Notably, the co-enrichment of Pathogenic Escherichia coli infection (pathogen-specific) and VEGF signaling pathway (host angiogenic) suggests unique dual-action antimicrobial/repair mechanisms. The Apoptosis-multiple species pathway conservation highlights evolutionary optimized targets for interspecies efficacy translation.

### 3.6. Combined Analysis

To elucidate the relationships among candidate metabolites, putative targets, diseases, and pathways, we constructed an integrative regulatory network (Figure 6). The analysis identified 9 metabolites associated with 27 diseases, which were further linked to 22 key targets from the PPI results and the top 20 enriched pathways. Based on Maximal Clique Centrality (MCC) scoring, the network prioritized 5 core metabolites (L-α-Glutamyl-L-Glutamic Acid, Maltitol, D-Maltose, Maltotriose, and Cytidine 5′-monophosphate), 5 pivotal targets (STAT3, HSP90AA1, CASP3, CASP8, and SRC), 3 disease modules (Infections, Neoplasms, Hyperphagia), and 2 key pathways (hsa05200: Pathways in cancer; hsa04217: Necroptosis).

### 3.7. Molecular Docking

Molecular docking was performed between the five screened core metabolites and five core targets. All binding energies of the docking results were below 0 and less than −5 kcal/mol (Figure 7A), indicating spontaneous and stable binding between ligands and receptors, thereby further validating the accuracy of the network pharmacology findings. Analysis of binding energies revealed that Maltitol exhibited the lowest binding energy with STAT3 (−7.10 kcal/mol), while Maltotriose demonstrated the most favorable binding energies with HSP90AA1 (−8.60 kcal/mol), CASP3 (−7.10 kcal/mol), CASP8 (−6.40 kcal/mol), and SRC (−8.10 kcal/mol), suggesting their superior binding stability.

The binding mode analysis demonstrated that Maltotriose interacted with HSP90AA1 (Figure 7C) and CASP3 (Figure 7D) through van der Waals forces, conventional hydrogen bonds, unfavorable donor-donor interactions, and carbon-hydrogen bonds. Similarly, Maltitol-STAT3 (Figure 7B) and Maltotriose-CASP8/SRC (Figure 7E,F) complexes-maintained stability via van der Waals interactions, conventional hydrogen bonds, and unfavorable donor-donor interactions. Collectively, these observations confirm that van der Waals forces, conventional hydrogen bonds, and unfavorable donor-donor interactions serve as critical stabilizing factors in ligand-receptor binding.

### 3.8. Metabolic Pathway Analysis of Key Metabolites

Based on network pharmacology and molecular docking screening, we identified two key metabolites, maltotriose and maltitol, for in-depth analysis. Both metabolites belong to carbohydrates, suggesting the potential significance of carbohydrates in *H. cordata*. Notably, maltotriose showed high accumulation in YN-cultivated *H. cordata*, whereas maltitol was predominantly accumulated in SC-cultivated *H. cordata*.

As illustrated in the metabolic pathway (Figure 8A), UDP-glucose is converted to amylose through waxy-mediated catalysis, subsequently forming starch via the enzymatic actions of GBE1/glgB. This starch is then hydrolyzed by alpha-amylase or beta-amylase to yield maltotriose, which undergoes hydrogenation to form maltitol. Transcriptomic analysis revealed 178 differentially expressed genes (DEGs) in this pathway, with their distinct expression patterns across geographical cultivation regions being visualized through heatmap analysis (Figure 8B).

Through correlation analysis (r > 0.7, *p* < 0.01), we constructed a regulatory network between these two metabolites and candidate DEGs (Figure 8C). Specifically, 10 genes demonstrated strong correlations with maltotriose: four encoding GBE1/glgB (Cluster-66,907.0, Cluster-72,962.0, Cluster-72,962.3, Cluster-72,962.5), three alpha-amylases (Cluster-12,112.0, Cluster-13,307.0, Cluster-63,294.2), and three beta-amylases (Cluster-67,616.4, Cluster-80,276.3, Cluster-80,276.7). Conversely, four genes showed significant correlations with maltitol: one GBE1/glgB (Cluster-72,962.4), two alpha-amylases (Cluster-74,155.2, Cluster-74,155.6), and one beta-amylase (Cluster-68,523.1). These findings suggest that the expression patterns of these genes may regulate the differential accumulation of these metabolites.

## 4. Discussion

*H. cordata*, as a dual-purpose medicinal and edible plant, holds significant importance in the development and utilization of its therapeutic value. Current research predominantly focuses on secondary metabolites such as flavonoids [2], terpenoids [33], and alkaloids [7]. However, its abundant nutritional components have been largely overlooked. Amino acids, fundamental units of protein structure, play pivotal roles in human nutrition and exhibit extensive pharmaceutical applications [34]. Nucleotides, constituting the basic units of nucleic acids (DNA and RNA), serve crucial functions in genetic information transmission and immune system regulation. These compounds are essential for immune cell proliferation and differentiation [35]. Vitamins are indispensable organic compounds that maintain normal physiological functions, participating critically in human growth, development, and metabolic processes, with distinct biological roles attributed to different vitamin types [36]. Steroidal compounds demonstrate substantial pharmacological potential in physiological regulation, with numerous steroidal drugs already achieving widespread clinical application [37].

Plant metabolites exhibit substantial compositional and quantitative variations due to genetic characteristics and growth environment disparities [38]. Leguminous plants such as soybeans and mung beans demonstrate rich amino acid profiles, particularly in lysine content [39], whereas cereal crops like wheat and rice maintain more comprehensive amino acid diversity [40]. Notably, spinach contains measurable γ-aminobutyric acid, which exhibits physiological activities including antihypertensive effects [41]. Environmental factors such as light intensity, temperature, and soil fertility significantly influence nucleotide levels [42]. Under optimal illumination, tomato leaves exhibit higher nucleotide content compared to low-light conditions, attributable to enhanced photosynthetic activity providing increased energy and precursors for nucleotide biosynthesis [43]. Steroidal saponins extracted from plants, exemplified by diosgenin from Dioscorea species, serve as crucial raw materials for synthesizing steroidal hormone drugs [44]. In nutraceutical applications, plant extracts rich in steroidal compounds, such as ginsenosides from Panax ginseng with demonstrated antioxidant, anti-fatigue, and immunomodulatory properties have been commercially developed [45]. Previous studies have systematically investigated these nutrients: Zhao et al. (2022) identified 93 amino acids and derivatives, 62 nucleotides and derivatives, 21 carbohydrates, and 21 vitamins across six major coarse cereals, while Castro-Alves et al. (2021) reported relatively limited sugar and amino acid quantities in *Anethum graveolens* [46,47]. In this study, we identified 264 amino acids and derivatives, 123 carbohydrates, 24 vitamins, 75 nucleotides and derivatives, and 10 steroids in *H. cordata* cultivated across six geographical regions. Compared with prior research in *H. cordata* documenting 92 amino acids and derivatives and 20 vitamins [2], our findings demonstrate substantial advancements in metabolite identification. This breakthrough establishes a critical foundation for investigating metabolite functionality and geographical variation mechanisms.

Beyond their nutritional contributions, these metabolites demonstrate substantial pharmacological significance. Through network pharmacology analysis, we identified nine candidate metabolites associated with multiple therapeutic targets and diseases, with enriched pathways linked to cancer and necroptosis. Molecular docking results revealed maltotriose and maltitol as key metabolites mediating region-specific pharmacological effects across different geographical cultivation sites. Previous investigations have established maltotriose’s critical associations with neoplasms, polyploidy, depressive disorder, and drug-related adverse reactions [48,49,50,51,52]. Maltitol, a polyol sugar alcohol, demonstrates limited yet notable inhibitory effects on periodontal and gingival inflammation [53,54]. Our findings further associate both metabolites with infections, neoplasms, and hyperphagia, showing strong binding affinities to STAT3, HSP90AA1, CASP3, CASP8, and SRC targets. Notably, maltotriose exhibited high accumulation in Yunnan-cultivated *H. cordata*, while maltitol predominantly accumulated in Sichuan-cultivated materials. Collectively, these findings establish crucial reference foundations for advancing investigations into its medicinal properties.

The pronounced geographical variations in key nutritional metabolites, particularly maltotriose and maltitol, revealed by this study offer a robust scientific foundation for implementing precision cultivation strategies tailored to specific end-uses of *H. cordata*. Our findings demonstrate that the cultivation region is a critical determinant of the plant’s phytochemical profile, significantly impacting its potential pharmaceutical and nutraceutical value. Sichuan cultivars (high maltitol): Plants cultivated in Sichuan (SC) exhibited significantly higher accumulation of maltitol. Given the predicted strong binding affinity of maltitol to key targets like STAT3 and its association with metabolic pathways (e.g., insulin signaling), SC-cultivated *H. cordata* emerges as a prime candidate for sourcing raw material targeted at developing products for metabolic health. This includes, but is not limited to, formulations aimed at managing diabetes and associated complications, or functional foods designed for glycemic control and metabolic syndrome management. Cultivation practices in Sichuan should be optimized to further enhance and stabilize maltitol production. Yunnan cultivars (high maltotriose): Conversely, *H. cordata* from Yunnan (YN) showed predominant enrichment of maltotriose. Molecular docking confirmed the exceptional binding stability of maltotriose with multiple therapeutic targets, including HSP90AA1, CASP3, CASP8, and SRC, which are pivotal in inflammation, apoptosis, and oncogenic signaling. This, coupled with the metabolite’s predicted association with anti-infective pathways, strongly positions YN-cultivated material as the optimal source for developing potent antiviral and anti-inflammatory formulations. Cultivation efforts in Yunnan should focus on maximizing maltotriose yield and consistency for these specific therapeutic applications.

To investigate the differential accumulation of these two critical metabolites, we analyzed their metabolic pathways. Maltotriose, primarily present during starch hydrolysis, serves as an intermediate product in starch saccharification. Starch is initially degraded into oligosaccharides such as maltose and maltotriose by enzymes including α-amylase and β-amylase, followed by further hydrolysis to glucose. Maltotriose, a trisaccharide composed of three glucose units linked via α-1,4-glycosidic bonds, is synthesized directly from starch [55]. In Saccharomyces cerevisiae, maltotriose biosynthesis shares regulatory pathways with maltose metabolism. Notably, maltose and maltotriose undergo distinct enzymatic hydrolysis: maltose is cleaved by maltase, whereas maltotriose is hydrolyzed by α-glucosidase [56]. In this study, we reconstructed the metabolic pathways of maltotriose and maltitol. UDP-glucose, a direct precursor for starch biosynthesis, plays a pivotal role in *H. cordata*. Specifically, UDP-glucose transfers glucose moieties to growing starch chains under the catalysis of AGPase (ADP-glucose pyrophosphorylase), forming amylose through α-1,4-glycosidic linkages [57]. Amylose, one of starch’s two primary components alongside amylopectin, constitutes linear glucose chains. During starch granule formation, amylose and amylopectin are synthesized concurrently. In contrast to the linear structure of amylose, amylopectin features branched configurations formed by α-1,6-glycosidic bonds connecting α-1,4-linked glucose chains [58]. Starch hydrolysis generates maltotriose through progressive cleavage of α-1,4-glycosidic bonds, releasing intermediate products including maltotriose and maltose. Subsequently, maltose and maltotriose are reduced to maltitol via hydrogenation [59].

Based on this metabolic pathway, we identified GBE1/glgB, α-amylases, and β-amylases as key candidate genes regulating the differential accumulation of maltotriose and maltitol. Studies across plant species support their roles: In cereals such as maize and rice, enhanced GBE1 activity increases starch branching degree, thereby impeding β-amylase’s ability to sequentially cleave α-1,4 glycosidic bonds and resulting in elevated maltotriose levels from incomplete hydrolysis [60,61]. Conversely, GBE1-deficient barley mutants exhibit reduced starch branching, enabling α-amylase to hydrolyze starch more efficiently and generate maltotriose [62]. High α-amylase activity in wheat accelerates starch degradation, producing substantial maltotriose [63]. In rice, OsAMY3 preferentially acts on long-chain amylose to yield maltotriose as an intermediate [64]. Potato tubers, characterized by low α-amylase activity, display minimal maltotriose accumulation [65]. During barley germination, β-amylase (e.g., HvBAM1) activity surges synergistically with α-amylase, though maltotriose levels remain contingent on starch branching patterns [66]. Maltitol biosynthesis primarily relies on hydrogenation processes. Collectively, maltotriose accumulation stems from two mechanisms: random α-amylase cleavage andβ-amylase inhibition by branched starch structures [67]. Plant species, enzymatic activity profiles, and starch architecture jointly determine its quantitative variation.

The findings provide a scientific foundation for the future industrial development of *H. cordata*. By elucidating the geographical variation in nutritional metabolites and their underlying regulatory mechanisms, this study offers critical insights for optimizing cultivation strategies. In the long term, this knowledge can be applied to practice precision agriculture by selecting the most suitable regions and agronomic practices for cultivating *H. cordata* with desired metabolite profiles, thereby improving its quality and value for specific nutraceutical applications. Regarding processing and product development, clarifying the medicinal value and biosynthetic mechanisms of critical metabolites could facilitate the creation of high-value-added products, such as novel pharmaceuticals, nutraceuticals, and functional foods. These advancements would address the growing demand for health-oriented products, promote diversified industrial development, enhance competitiveness, and stimulate regional economic growth.

While this study has elucidated the influence of geographical environments on metabolite accumulation in *H. cordata*, the interactions among environmental factors remain incompletely disentangled. Although network pharmacology and molecular docking have predicted potential targets and pathways of key metabolites, direct functional validation through in vitro or in vivo experiments is still lacking. Transcriptomic analysis identified differentially expressed genes (DEGs); however, their direct regulatory relationships with metabolite biosynthesis remain uncharacterized. Future investigations should prioritize these research gaps to advance mechanistic understanding. Importantly, the current findings on geographical metabolite variations provide an immediate foundation for implementing precision cultivation strategies.

Beyond environmental factors, the stability of the underlying genotype and its interaction with the environment (G × E) constitute a second layer of complexity that must be resolved before region-specific cultivation can be commercialized with confidence. The present metabolite maps were generated with plant materials collected from six provinces, yet all accessions were of the same landrace; consequently, we cannot discern whether the observed maltitol–maltotriose divergence is driven primarily by environment, by cryptic genetic divergence among regions, or by G × E. If a grower in Sichuan adopts a “diabetes-targeted” genotype selected for high maltitol only to find that the same seed lot produces low maltitol when cultivated in a different year or at a slightly higher elevation, product efficacy—and regulatory compliance—will be compromised. Multi-year, multi-location reciprocal transplant trials coupled with high-density SNP genotyping are therefore urgently needed to (i) estimate broad-sense heritability (H^2^) and genotype-by-environment variance components for maltitol and maltotriose, (ii) identify stable “core” genotypes that maintain therapeutic metabolite ratios across divergent environments, and (iii) develop PCR-based markers (e.g., GBE1/glgB haplotypes, α-amylase copy-number variants) that breeders can use to track metabolite stability during seed multiplication. Only after G × E patterns are quantified can region-specific branding (e.g., “SC maltitol-type” or “YN maltotriose-type”) be guaranteed to deliver consistent medicinal value to end-users.

## 5. Conclusions

This study systematically characterized the geographical variations in nutritional metabolites of *H. cordata* cultivated across six Chinese regions, integrating metabolomic, transcriptomic, and pharmacological approaches. A total of 496 metabolites were identified, with maltotriose and maltitol emerging as key metabolites. Network pharmacology prioritized STAT3, HSP90AA1, CASP3, CASP8, and SRC as core therapeutic targets, with molecular docking confirming stable interactions between these two metabolites and targets, highlighting their potential in managing infections, neoplasms, and metabolic disorders. Transcriptomic analysis revealed that the differential expression of starch metabolism-related genes, including GBE1/glgB, α-amylases, and β-amylases, likely drives these geographical disparities. These results not only elucidate the molecular mechanisms underlying *H. cordata*’s pharmacological potential but also establish a foundation for precision agriculture and industrial applications. Future research should focus on the functional validation of candidate genes and in vivo studies to confirm therapeutic efficacy. By bridging geographical, metabolic, and pharmacological insights, this work positions *H. cordata* as a promising candidate for developing natural health products.

## Figures and Tables

**Figure 1 metabolites-15-00701-f001:**
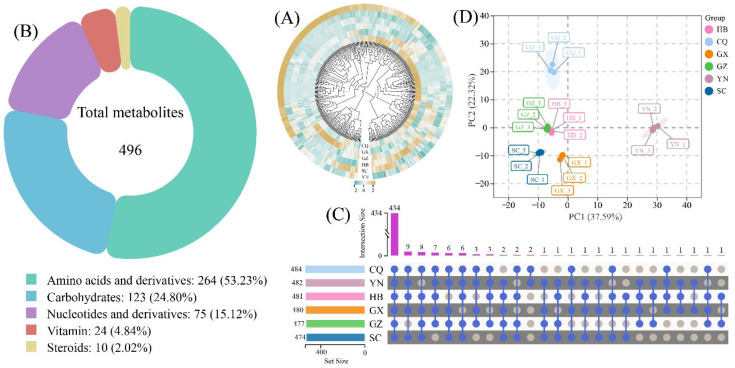
Comprehensive characterization of nutritional metabolites identified in *H. cordata* cultivated across distinct geographical regions. (**A**) Heatmap visualization of the identified metabolites. (**B**) Taxonomic classification of the identified metabolites. (**C**) Quantitative analysis of metabolite counts, overlapping metabolites, and region-specific metabolites. (**D**) Principal component analysis (PCA) based on the identified metabolite profiles. Blue circles, presence of the feature in the corresponding sample; gray circles, absence.

**Figure 2 metabolites-15-00701-f002:**
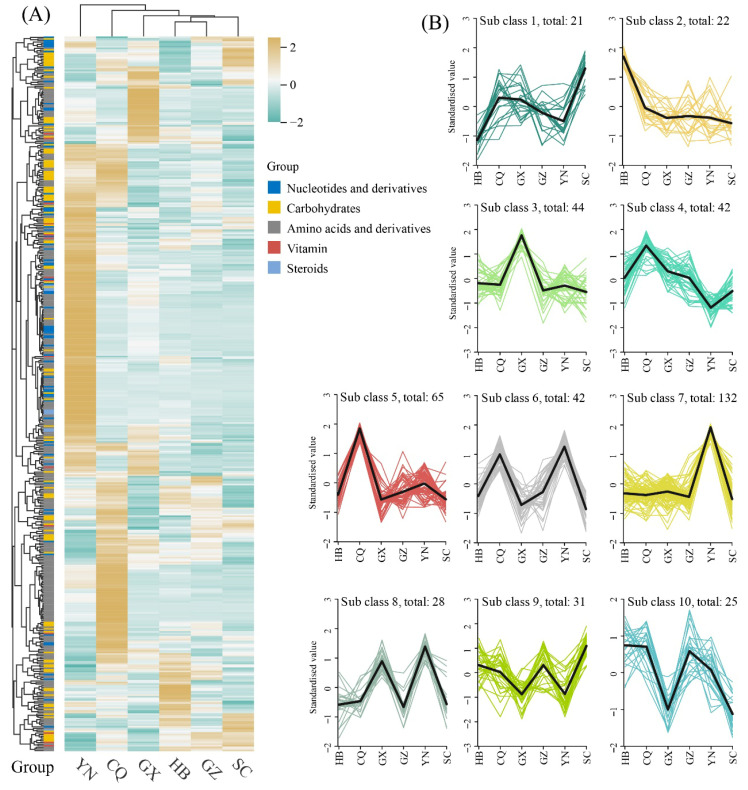
Systematic evaluation of differentially accumulated metabolites (DAMs) in *H. cordata* from diverse geographical regions. (**A**) Heatmap representation of DAMs. (**B**) Cluster pattern analysis of DAMs using K-means algorithm.

**Figure 3 metabolites-15-00701-f003:**
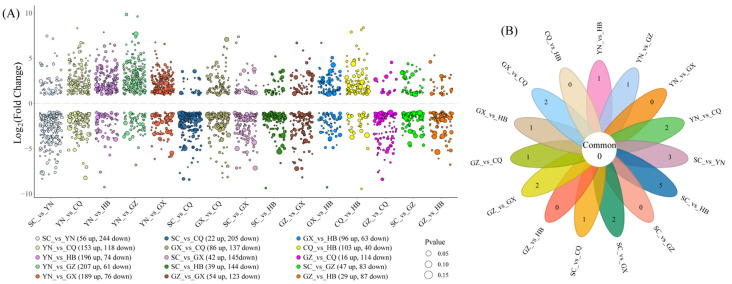
Cross-comparative investigation of differentially accumulated metabolites among different comparison groups. (**A**) Up- and down-regulated accumulation patterns of DAMs across comparison groups. (**B**) Petal Venn diagram illustrating overlaps and unique DAMs among comparison groups.

**Figure 4 metabolites-15-00701-f004:**
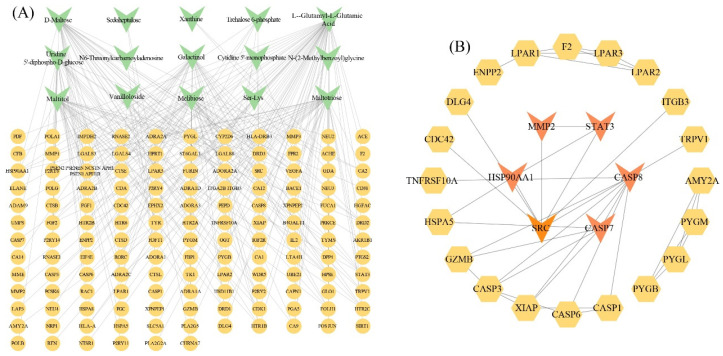
Analysis of putative targets for metabolites in *H. cordata* cultivated in different geographical regions. (**A**) Interaction network between putative targets and metabolites. (**B**) Protein–protein interaction (PPI) network of putative targets. Green shapes represent metabolites and orange-yellow shapes represent putative targets.

**Figure 5 metabolites-15-00701-f005:**
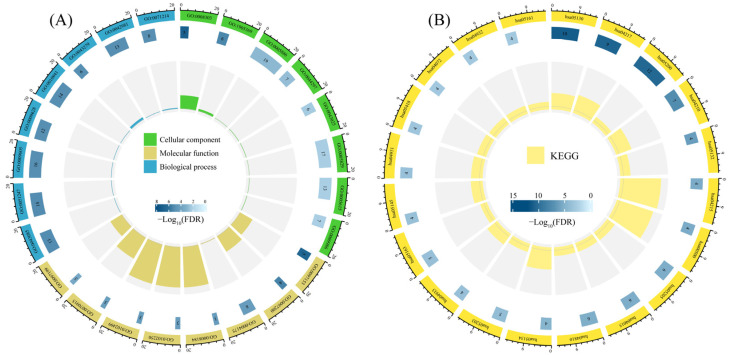
Enrichment analysis of candidate targets for metabolites. (**A**) Gene Ontology (GO) enrichment analysis. (**B**) Kyoto Encyclopedia of Genes and Genomes (KEGG) pathway enrichment analysis.

**Figure 6 metabolites-15-00701-f006:**
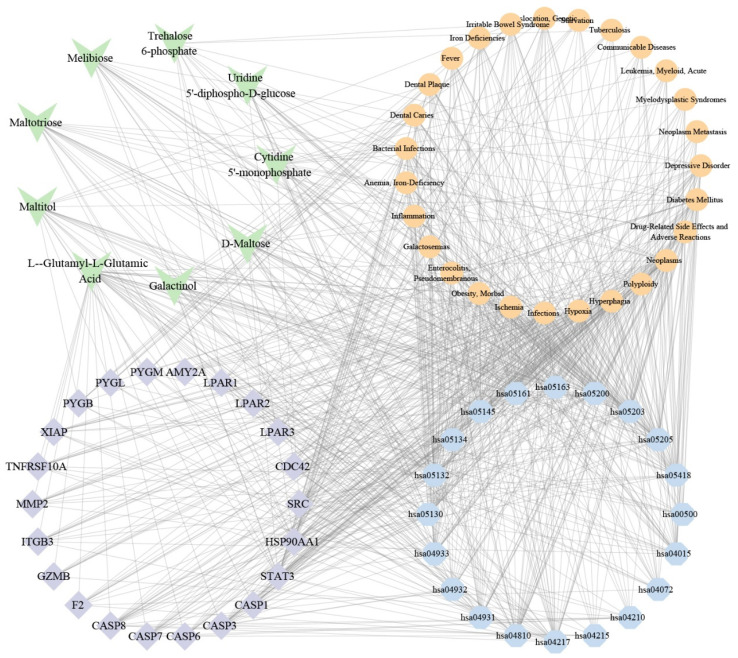
Interactive network mapping of functional metabolite-target-pathway-efficacy correlations. Green shapes represent metabolites. Purple shapes represent putative targets. Blue shapes represent the top enriched KEGG pathways, and orange shapes represent associated disease modules. Edges represent interactions between the components.

**Figure 7 metabolites-15-00701-f007:**
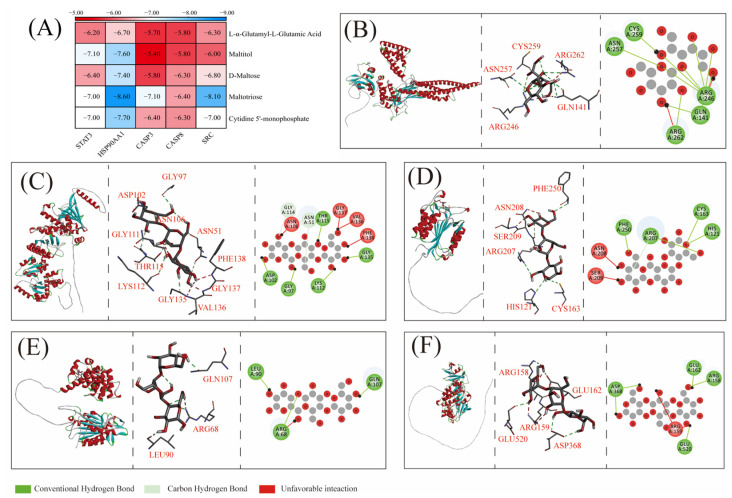
Molecular docking conformation. (**A**) Heatmap of binding energies (kcal/mol) for the five core metabolites (L-α-Glutamyl-L-Glutamic Acid, Maltitol, D-Maltose, Maltotriose, and Cytidine 5′-monophosphate) against the five targets (STAT3, HSP90AA1, CASP3, CASP8, SRC). Detailed 3D visualization of the molecular docking conformations for the most stable complexes: (**B**) Maltitol with STAT3, Maltotriose with HSP90AA1 (**C**), CASP3 (**D**), CASP8 (**E**), SRC (**F**).

**Figure 8 metabolites-15-00701-f008:**
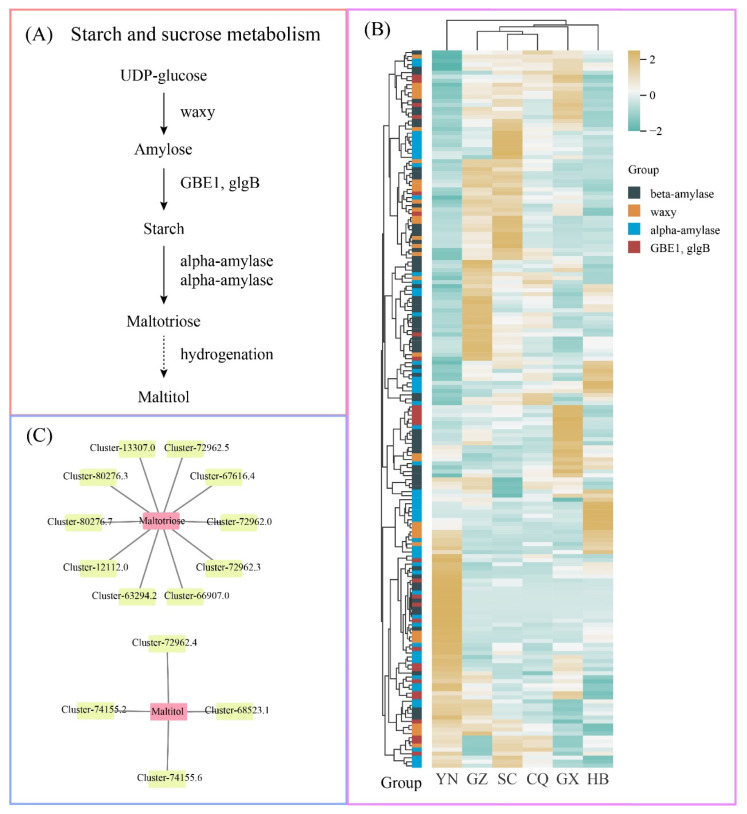
Metabolic pathway analysis of key metabolites. (**A**) Metabolic pathway analysis of maltotriose and maltitol. (**B**) Analysis of differentially expressed genes in metabolic pathways. (**C**) Regulatory network diagram between key metabolites and candidate differentially expressed genes.

## Data Availability

The data that support the findings of this study are available from the corresponding author upon reasonable request.

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
