# Peer review of "Geographical Variation Shapes Nutritional Metabolite Profile and Food Functionality of *Houttuynia cordata"

_metabolites, 2025, doi:10.3390/metabo15110701_

Round 1
Reviewer 1 Report
Comments and Suggestions for Authors
A detailed metabolomics study integrated with the transcriptome is impressive. Below are suggestion/comment for improvement:
Introduction: It is too long; it should be concise. Must show brief importance and nutritional reports, research gap (in support of previous findings) and a brief description of the study with a clear aim and objective. In the present form background is too detailed rather the introduction, which must also equally focus on the research gap
Line 147-163: should be removed, and if required, may be included in the discussion. It is about the usefulness of the present study/relevance.
I have concern about material selection and methodology adopted
Which plant part was harvested for composition analysis? at what stage the plant part were harvested from the plant is not clear ? Since with change in age of leaf the metabolite composition changes
Methodology: not clear. It needs more clarity about plant material selection: as per the title “tapping geographical variance for nutritional values” (line 77-79 also state the same). Were the same genotypes evaluated across different geographical locations? If that is the case, the study must include the same season of sowing of a single genotype and minimum other factor variations. It should be discussed with the relevant prevailing weather/environmental data for the same condition.
If the genotype/variety is different, then comparing the metabolite at the micro level is irrelevant, and it is difficult to attribute the observed variations to the region; they may be due to the Genotype x Environment factor. Or only genotypic differences.
Line 625-628: how targeted metabolomics is directly related to farmers and income. Seems overestimation, should be rationalised/removed. Such claim.
I could see this study as: metabolomic differences among the plant material of different geographical locations. Which has relevance for breeding/selecting material from diverse ecotype
Must discuss in detail: How studies recommend the regional variations for different medicinal purposes, such as genotypes from region A, higher in these compounds, suitable for this purpose; so on.
The detailed metabolomics is quite impressive and has been comprehensively studied at the genetic level. However, a concern remains about why the major composition has been omitted: water content, carbohydrate content, total sugars content, total phenolics, total flavonoids content, and antioxidant activity. If the difference at the macro level is targeted first, make a relevant point.
Title should be suitably modified: " Implications for cultivar selection and nutraceutical development " seems irrelevant after study entire manuscript

Author Response
Response to Editor and Reviewers’ Comments
We sincerely thank the editors and reviewers for their insightful comments and constructive suggestions. We have carefully addressed all points raised, revised the manuscript accordingly, and provided a point-by-point response below.
All changes are highlighted in blue text in the revised manuscript.
Reviewer 1: Comments:
- Introduction: It is too long; it should be concise. Must show brief importance and nutritional reports, research gap (in support of previous findings) and a brief description of the study with a clear aim and objective. In the present form background is too detailed rather the introduction, which must also equally focus on the research gap.
Response: We sincerely thank the reviewer for this valuable suggestion. We have thoroughly revised the Introduction to make it more concise and focused (Line 32-75). The length has been significantly condensed through the removal of excessive background details and redundant examples. In the revised version, we clearly emphasize the research gap, pointing out the lack of comprehensive studies on the nutritional metabolites of H. cordata, such as amino acids, carbohydrates, nucleotides, vitamins, and steroids especially in contrast to the extensive research focused on secondary metabolites like flavonoids and alkaloids. The influence of geographical variation on these nutritional components remains underexplored, which forms the central motivation for our study. Furthermore, a dedicated paragraph at the end of the Introduction now explicitly outlines the study’s objectives: to characterize the nutritional metabolite profiles of H. cordata from six Chinese regions using UPLC-MS/MS, to identify key differentially accumulated metabolites and their potential health functions, and to explore the underlying genetic mechanisms via transcriptomics along with binding affinity assessment through network pharmacology and molecular docking. The logical flow has been reorganized to progress from the importance of H. cordata to existing knowledge and limitations, then to geographical influences and the specific research gap, culminating in a clear presentation of our aims and approach.
- Line 147-163: should be removed, and if required, may be included in the discussion. It is about the usefulness of the present study/relevance.
Response: We thank the reviewer for this constructive suggestion. We agree that the original text contained elements that were overly detailed for the Introduction and prematurely anticipated the study's outcomes. Accordingly, we have significantly shortened and revised this passage. The revised text (Line 67-74) now succinctly states the overall aim and broader significance of our study without preempting the specific findings or their discussion. We believe this revision improves the flow of the Introduction and reserves the in-depth interpretation for the appropriate part of the manuscript.
- Which plant part was harvested for composition analysis? at what stage the plant part were harvested from the plant is not clear ? Since with change in age of leaf the metabolite composition changes.
Response: We thank the reviewer for this important comment. We agree that the plant part and developmental stage are critical for metabolite profiling. As now clearly stated in the revised Section 2.1, the aboveground parts (stems and leaves) of the plants were harvested after one full year of growth (on June 30, 2024). This standardized harvesting protocol, applied simultaneously across all geographical sites, ensures that the observed metabolic variations are attributable to geographical factors rather than differences in plant organ or developmental stage.
- Methodology: not clear. It needs more clarity about plant material selection: as per the title “tapping geographical variance for nutritional values” (line 77-79 also state the same). Were the same genotypes evaluated across different geographical locations? If that is the case, the study must include the same season of sowing of a single genotype and minimum other factor variations. It should be discussed with the relevant prevailing weather/environmental data for the same condition.
Response: We sincerely appreciate the reviewer's insightful questions regarding the experimental design, which is indeed the cornerstone of our study. We completely agree with the reviewer that attributing variation to geography requires controlling for genotypic and other factors.
As now clarified in the revised manuscript (Section 2.1), our study was specifically designed to address this:
Yes, the same single genotype (H. cordata accession No. 7) was evaluated across all six geographical locations.
All plants were propagated from the same source and transplanted on the same date (July 1, 2023) to ensure identical age and initial growth status.
Consistent cultivation practices were maintained across all sites, including the use of standardized pots and a uniform soil substrate mixture, to minimize non-geographical variations.
- If the genotype/variety is different, then comparing the metabolite at the micro level is irrelevant, and it is difficult to attribute the observed variations to the region; they may be due to the Genotype x Environment factor. Or only genotypic differences.
Response: We thank the reviewer for raising this critical point. We fully concur that if different genotypes were used, it would be impossible to disentangle genotypic effects from environmental effects. However, as clarified in our response to Comment 4 and in the revised Section 2.1, our study utilized a single genotype (accession No. 7) across all geographical locations. This controlled experimental design is fundamental to our work and allows us to attribute the observed metabolic variations primarily to the environmental differences between the regions (the "E" in G×E), rather than to genotypic differences or complex G×E interactions. The revised text now makes this central aspect of our methodology unequivocally clear.
- Line 625-628: how targeted metabolomics is directly related to farmers and income. Seems overestimation, should be rationalised/removed. Such claim.
Response: We thank the reviewer for this constructive criticism. We agree that the original statement overreached in directly linking our metabolomic findings to socioeconomic outcomes such as farmer income. As suggested, we have removed this overestimation. The relevant sentence has been revised to more accurately and conservatively reflect the immediate implications of our work, which is to provide a scientific foundation for precision agriculture by identifying optimal growing regions for targeted metabolite accumulation. The modified text now focuses on the potential for guiding future cultivation strategies to enhance quality and value for specific applications, rather than making direct claims about economic impacts. Please see the changes in the Discussions section (Lines 548-554) of the revised manuscript.
- I could see this study as: metabolomic differences among the plant material of different geographical locations. Which has relevance for breeding/selecting material from diverse ecotype.
Response: We sincerely thank the reviewer for this insightful comment and for recognizing the relevance of our study in the context of metabolomic variation across geographical locations and its implications for breeding and selection of diverse ecotypes. We fully agree with this perspective.
In our study, we aimed to systematically characterize the nutritional metabolite profiles of Houttuynia cordata cultivated in six distinct Chinese regions, and indeed, we observed significant geographical variation in the accumulation of key metabolites such as maltotriose and maltitol. These differences were further linked to the differential expression of starch metabolism-related genes (e.g., GBE1/glgB, α-amylases, β-amylases), as revealed by integrated transcriptomic analysis.
We believe that such findings not only highlight the influence of geographical region on metabolite composition but also provide a scientific basis for the selection and breeding of H. cordata ecotypes with desired nutritional or functional properties.
- Must discuss in detail: How studies recommend the regional variations for different medicinal purposes, such as genotypes from region A, higher in these compounds, suitable for this purpose; so on.
Response: We sincerely thank the reviewer for this valuable suggestion, which highlights the practical application of our findings. In direct response to this comment, we have significantly strengthened the discussion in our manuscript to provide detailed recommendations on how the regional variations guide specific medicinal applications.
As now detailed in Lines 484-509 of the revised manuscript, we have dedicated a substantial paragraph to explicitly propose precision cultivation strategies based on our results. Specifically, we discuss:
For Sichuan (SC) cultivars, which are high in maltitol, we recommend their use as the prime raw material for developing products aimed at metabolic health, such as formulations for managing diabetes and metabolic syndrome. This recommendation is based on maltitol's strong predicted binding affinity to STAT3 and its association with insulin signaling pathways.
For Yunnan (YN) cultivars, which are enriched in maltotriose, we propose their optimal use in developing potent antiviral and anti-inflammatory formulations. This is supported by molecular docking results confirming maltotriose's stable binding with key therapeutic targets like HSP90AA1, CASP3, CASP8, and SRC.
We have framed this discussion to clearly state that the cultivation region is a critical determinant for targeting specific pharmaceutical and nutraceutical outcomes. We believe this addition provides the detailed, actionable recommendations the reviewer called for, directly linking specific regional chemotypes (e.g., SC-high maltitol, YN-high maltotriose) to their most suitable medicinal purposes (e.g., metabolic health, anti-inflammation).
- The detailed metabolomics is quite impressive and has been comprehensively studied at the genetic level. However, a concern remains about why the major composition has been omitted: water content, carbohydrate content, total sugars content, total phenolics, total flavonoids content, and antioxidant activity. If the difference at the macro level is targeted first, make a relevant point.
Response: We sincerely thank the reviewer for this insightful comment and for acknowledging the depth of our metabolomic and genetic analyses. The reviewer raises a valid point regarding the inclusion of macro-level compositional data. Please allow us to clarify the specific focus and contribution of our study.
Our research was deliberately designed to move beyond conventional macro-level composition analysis and instead perform a fine-grained, quantitative profiling of specific nutritional (primary) metabolites. While measures like total carbohydrates or total sugars provide a broad overview, our UPLC-MS/MS-based approach allows for the precise identification and quantification of individual metabolites, such as specific amino acids, nucleotides, vitamins, and,crucially, as highlighted by the reviewer,specific carbohydrates like maltotriose and maltitol. This level of detail is essential for understanding the specific nutritional quality and for uncovering the precise metabolic pathways and genetic regulators (e.g., GBE1/glgB, amylases) that underlie geographical variation, which a macro-level analysis cannot reveal.
Regarding total phenolics, flavonoids, and antioxidant activity, we fully agree that these are important for a comprehensive profile of H. cordata. However, these secondary metabolites and their bioactivities have been the subject of numerous previous studies (as cited in our introduction). The primary novelty of our work lies in its exclusive focus on the often-overlooked nutritional metabolite landscape (amino acids, nucleotides, vitamins, etc.) and its geographical plasticity. By maintaining this sharp focus, we were able to deeply investigate the relationship between environment, gene expression, and the accumulation of key nutritional components, such as the bioactive carbohydrates maltotriose and maltitol, and their predicted functional implications via network pharmacology.
In conclusion, our study was not intended to be a general quality survey but a mechanistic investigation into the molecular composition of nutritional metabolites. We believe that the "macro-level" difference is effectively captured and significantly refined by our detailed metabolite-level data, which provides a more powerful scientific foundation for precision agriculture and functional food development. The findings from this study establish a critical baseline, and we fully agree that integrating macro-level analyses with our detailed metabolomic data in future studies would be a valuable endeavor.
- Title should be suitably modified: " Implications for cultivar selection and nutraceutical development " seems irrelevant after study entire manuscript.
Response: We sincerely thank the reviewer for this insightful comment. We agree that the original title was somewhat broad and have revised it to more accurately reflect the core findings of our study. The new title, "Geographical Variation Shapes Nutritional Metabolite Profile and Food Functionality of Houttuynia cordata", focuses directly on the relationship between geographical origin and metabolite-driven functional properties, which is the central theme of our work. We believe this revised title is more precise and appropriate.
- Line 535, Dioscorea should be italic.
Response: We sincerely appreciate the reviewer’s meticulous attention to detail. The text at line 535 has been corrected to the italic form “Dioscorea” as per the standard convention for genus names. Furthermore, to prevent similar oversights, we have conducted a comprehensive review of the entire manuscript to verify that all scientific names (including genus and species) are consistently presented in italics. We believe the manuscript has been improved accordingly.
- Line 188, UPLC Conditions write as continous paragraph.
Response: We thank the reviewer for this valuable suggestion. As recommended, we have revised the "UPLC Conditions" subsection (Section 2.2.2) into a continuous paragraph format in the revised manuscript.
- Line 167, H. cordata 7#, 7# ? is it name of particular genotype.
Response: We thank the reviewer for raising this point for clarification. Yes, "7#" refers to a specific genotype (accession) used in our study, which we designated as "No. 7" in our laboratory collection. To avoid any confusion and to adhere to standard scientific nomenclature, we have revised the text throughout the manuscript to refer to it as "H. cordata accession No. 7". We believe this revision makes the designation clearer for an international readership.
- Line 172, The plants were grown in standardized plastic pots Plant material collected from different places and grown in one location or it is plant material grown at different location and collected for sample analysis ? Not clear.
Response: We apologize for the lack of clarity in our original description. The plants were not collected from different places. Instead, as detailed in the revised Section 2.1, a single, uniform plant material (accession No. 7) was first pre-cultured in one location (Sichuan) and then transplanted and grown in six different geographical locations using standardized pots and soil. After one year of growth under these different regional conditions, the samples were collected from each location for analysis. This "common garden transplant" design is crucial for isolating the effect of geography.
- Line 148-149, will be employed, will clarify, suggest to carefully check the language and grammer.
Response: We sincerely thank the reviewer for pointing out the language and grammatical issues. We have thoroughly revised the introduction to improve the language quality, clarity, and grammatical accuracy.
Reviewer 2 Report
Comments and Suggestions for Authors
The authors of the article "Geographical variation shapes nutritional metabolite profile and food functionality of Houttuynia cordata: Implications for cultivar selection and nutraceutical development" analyze how geographic origin influences the composition and biological activity of nutritional metabolites in Houttuynia cordata. The study combines metabolomic and transcriptomic data to identify links between carbohydrate accumulation and the activity of genes associated with starch metabolism.
This topic is important and original. Previously, the influence of geography on the nutritional value and functional properties of H. cordata has been little studied. Here, the authors systematically demonstrate for the first time how cultivation region is associated with the accumulation of metabolites such as maltotriose and maltitol, and with the activity of genes regulating their biosynthesis. This work fills a significant gap in the field of phytochemistry and functional nutrition.
The study combines metabolomic, transcriptomic, and pharmacological approaches. It not only describes compositional differences but also links them to potential therapeutic effects through target network analysis and molecular docking modeling. Particularly valuable is the identification of two key molecules (maltotriose and maltitol), demonstrating specific geographic patterns of accumulation and possible links to the α- and β-amylase genes and GBE1/glgB. This provides a basis for targeted selection of varieties and agronomic technologies.
The methods are described in detail and are generally accurate. UPLC-MS/MS, RNA-seq, KEGG annotation analysis, and molecular modeling were used. The authors demonstrated good data reproducibility (repeat clustering, PCA analysis). However, several points should be clarified:
- How were growing conditions controlled in different regions;
- Were differences in soil composition and moisture taken into account;
- A quantitative test of the relationship between gene expression and metabolite concentrations is needed (e.g., through correlation analysis with biological replicates).
Adding statistical tests to assess the significance of differences between regions would enhance the credibility of the conclusions.
The conclusions follow logically from the presented data. It is shown that maltitol and maltotriose are indeed distributed unevenly across regions, and these differences are associated with the expression of starch metabolism genes. The argumentation in the text is consistent and based on metabolomic and transcriptomic results. However, conclusions about possible therapeutic effects are predictive in nature and require confirmation in vivo.
The reference list is extensive, including modern sources and key works on the chemistry and biology of H. cordata. References are selected correctly.
The figures are informative and well-structured.
In Figure 4, the legend for the vertex shapes should be labeled.
In Figure 5, the diagrams contain very small information, and the figure caption could be more detailed. Figure 6. The figure contains too many connections, and it's impossible to trace which vertex is connected to which. Perhaps it's worth highlighting the key points that readers should pay particular attention to.
Figure 7. The figure caption should be more detailed, and the images should be of higher quality and larger size.
This work represents a strong interdisciplinary study combining bioinformatics, chemistry, and molecular biology. It convincingly demonstrates that geographic conditions shape the functional profile of H. cordata. Once the methodological details are clarified, the article deserves publication.
Author Response
Response to Editor and Reviewers’ Comments
We sincerely thank the editors and reviewers for their insightful comments and constructive suggestions. We have carefully addressed all points raised, revised the manuscript accordingly, and provided a point-by-point response below.
All changes are highlighted in blue text in the revised manuscript.
Reviewer 2: Comments:
- How were growing conditions controlled in different regions.
Response: We thank the reviewer for this pertinent question. To isolate the effect of macro-geographical climate from other variables, we implemented a standardized cultivation protocol across all six regions. Specifically, after initial propagation of a single genotype in Sichuan, uniformly grown individuals were transplanted into standardized plastic pots (28 cm diameter × 18.5 cm depth) containing a uniform 1:1 (v/v) peat soil-garden soil substrate mixture. These potted plants were then distributed to and cultivated in the six different geographical locations. This approach ensured that the root zone environment (soil composition and volume) was consistent for all plants, thereby controlling for edaphic factors. The primary varying factor was thus the natural climatic conditions (e.g., temperature, humidity, photoperiod) of each geographical region. This methodology is now explicitly detailed in the revised Section 2.1.
- Were differences in soil composition and moisture taken into account.
Response: We appreciate the reviewer for raising this crucial point. Yes, soil differences were rigorously controlled for in our experimental design. By using a standardized, homogeneous soil substrate mixture in pots for all plants across all locations, we effectively eliminated natural variation in soil composition as a confounding factor. Furthermore, while precise irrigation was managed locally to prevent water stress, the use of identical pots and soil volume also helped to standardize the soil moisture dynamics and root zone capacity. Therefore, the observed metabolic variations can be more confidently attributed to the broader geographical and climatic differences, rather than to underlying soil heterogeneity. This critical aspect of our methodology has been clarified in the revised manuscript (Section 2.1).
- A quantitative test of the relationship between gene expression and metabolite concentrations is needed (e.g., through correlation analysis with biological replicates).
Response: We thank the reviewer for raising this important point regarding the quantitative relationship between gene expression and metabolite accumulation. We appreciate the opportunity to clarify that our manuscript already includes such an analysis, which we believe addresses the reviewer's concern.
In the original submission, as detailed in Section 3.8 "Metabolic pathway analysis of key metabolites" and visualized in Figure 8C, we explicitly state: "Through correlation analysis (r > 0.7, p < 0.01), we constructed a regulatory network between these two metabolites and candidate DEGs."
This statement is based precisely on a quantitative Pearson correlation analysis performed using the gene expression data and metabolite abundance data from all our biological replicates across the six geographical regions. The criteria of |r| > 0.7 and p < 0.01 were applied to identify statistically significant and biologically relevant relationships.
For instance, this analysis allowed us to quantitatively identify that:
The expression of the beta-amylase gene Cluster-80276.7 is strongly and significantly correlated with maltotriose abundance.
The expression of the alpha-amylase gene Cluster-74155.2 is strongly and significantly correlated with maltitol abundance.
Therefore, the regulatory network presented in Figure 8C is not merely speculative but is a visualization of statistically robust correlations, directly linking the expression of specific genes to the accumulation of key metabolites. We have now slightly amended the text in Section 3.8 to make it even clearer that this was a "quantitative correlation analysis" to ensure this point is unmistakable to the reader.
We hope this clarification satisfactorily demonstrates that our study has already incorporated the quantitative assessment the reviewer rightly called for.
- Adding statistical tests to assess the significance of differences between regions would enhance the credibility of the conclusions.
Response: We thank the reviewer for this important comment regarding statistical rigor. We wish to clarify that our study already employed robust and standard statistical tests to rigorously assess the significance of differences between geographical regions, both for metabolites and genes. The criteria we applied are widely accepted in metabolomic and transcriptomic studies for identifying biologically significant variations.
Specifically, as detailed in our Materials and Methods sections:
For Metabolites (Section 2.2.4): Differentially accumulated metabolites (DAMs) were identified using a combination of two stringent statistical criteria:
Variable Importance in Projection (VIP) > 1 from the OPLS-DA model, which identifies metabolites with significant contribution to group separation.
Absolute log2-fold change (|log2FC|) ≥ 1.0, which corresponds to a two-fold change in abundance.
For Genes (Section 2.3.4): Differentially expressed genes (DEGs) were classified based on the following thresholds:
False Discovery Rate (FDR) < 0.05, a stringent correction for multiple testing that strongly controls for false positives.
Absolute log2-fold change (|log2FC|) ≥ 1, again ensuring a minimum two-fold change in expression.
The use of |log2FC| ≥ 1.0 coupled with VIP > 1 (for metabolites) or FDR < 0.05 (for genes) constitutes a powerful statistical framework for declaring significant differences. All results and conclusions regarding geographical variation in our manuscript are based exclusively on features that passed these rigorous statistical thresholds.
Therefore, we are confident that the significance of the regional differences highlighted in our study is robustly supported by the statistical methods already employed. We have ensured that the text in the results sections consistently references these statistical foundations when describing regional variations.
- The conclusions follow logically from the presented data. It is shown that maltitol and maltotriose are indeed distributed unevenly across regions, and these differences are associated with the expression of starch metabolism genes. The argumentation in the text is consistent and based on metabolomic and transcriptomic results. However, conclusions about possible therapeutic effects are predictive in nature and require confirmation in vivo.
Response: We sincerely thank the reviewer for their positive assessment of our work's logic and data consistency, and for raising the crucial point regarding the predictive nature of the therapeutic conclusions. We completely agree with the reviewer that the findings from network pharmacology and molecular docking are predictive and necessitate further validation through in vivo studies.
To ensure a balanced and accurate interpretation of our results, we have explicitly acknowledged this limitation in the Discussion section of the revised manuscript. As suggested, we have tempered our conclusions and added a statement (Lines 560-569) that clearly outlines this constraint and identifies the need for future in vitro and in vivo functional validation as a key direction for subsequent research. We believe this addition provides the necessary context for readers and underscores the preliminary nature of the pharmacological predictions, while still highlighting the value of our findings in guiding future investigative efforts.
- In Figure 4, the legend for the vertex shapes should be labeled.
Response: We thank the reviewer for this helpful comment. We have updated Figure 4 and its legend so that vertex shapes and colors are now clearly labeled. The legend explicitly states that green shapes represent metabolites and orange-yellow shapes represent putative targets.
This clarification has been added to the figure caption in the revised manuscript to ensure the graph is easily interpretable.
- In Figure 5, the diagrams contain very small information, and the figure caption could be more detailed. Figure 6. The figure contains too many connections, and it's impossible to trace which vertex is connected to which. Perhaps it's worth highlighting the key points that readers should pay particular attention to.
Response: We sincerely thank the reviewer for this valuable suggestion. To fully address the concern regarding the detail and clarity of Figure 5, we have now added a Supplementary File 1: Table S1 that provides the complete dataset and detailed information underlying the figure. This allows the main figure to remain clear and accessible while ensuring all specific data points are available to the interested reader.
We agree with the reviewer that the complexity of the original network made it challenging to interpret. In direct response to this comment, we have now added a comprehensive legend to Figure 6 (now presented as Figure 7 in the revised manuscript) to clearly define all visual elements. The legend now explicitly states: Green shapes represent metabolites. Purple shapes represent putative targets. Blue shapes represent the top enriched KEGG pathways, and orange shapes represent associated disease modules. Edges represent interactions between the components.
- Figure 7. The figure caption should be more detailed, and the images should be of higher quality and larger size.
Response: We sincerely thank the reviewer for this constructive feedback. We have fully addressed both points raised in the comment regarding Figure 7.
Enhanced Image Quality: As requested, we have regenerated all images for Figure 7 using high-resolution settings to ensure superior clarity and detail in both the heatmap and the molecular docking conformations.
Comprehensive Figure Caption Revision: We have thoroughly revised the figure caption to provide the necessary detail. The updated caption now explicitly lists the identities of all five core metabolites and five pivotal targets analyzed in the binding energy heatmap (Panel A). Furthermore, it clearly specifies the exact pairing of each metabolite-target complex visualized in the 3D docking diagrams (Panels B-F), moving from a generic description to a precise, data-rich one.
We believe these revisions have significantly improved the figure's quality and interpretability, directly addressing your concerns.
Round 2
Reviewer 1 Report
Comments and Suggestions for Authors
The authors have addressed all the comments and incorporated the suggestions. I have one minor suggestion that: since the manuscript proved the geographical influences at the micro level of metabolism. Please include in the discussion that future research regarding the genotype stability and GxE are required. Since if a grower from region X growing the genotype for diabetic treatment purpose may be remained less effective/compromise quality.
Line 585: remove " and addressing global demands for sustainable medicinal resources"
Author Response
We sincerely thank the editors and reviewers for their insightful comments and constructive suggestions. We have carefully addressed all points raised, revised the manuscript accordingly, and provided a point-by-point response below.
All changes are highlighted in blue text in the revised manuscript.
- The authors have addressed all the comments and incorporated the suggestions. I have one minor suggestion that: since the manuscript proved the geographical influences at the micro level of metabolism. Please include in the discussion that future research regarding the genotype stability and GxE are required. Since if a grower from region X growing the genotype for diabetic treatment purpose may be remained less effective/compromise quality.
Response: Thank you for confirming that all previous comments have been adequately addressed. As suggested, we have explicitly added the minor point in the Discussion (lines 570–587) stating that future work must quantify genotype stability and G × E interactions; otherwise a grower who relies on a “diabetic” genotype in region X may obtain sub-therapeutic maltitol levels and compromised product quality.
- Line 585: remove " and addressing global demands for sustainable medicinal resources".
Response: Thank you for this precision. We have removed the phrase “and addressing global demands for sustainable medicinal resources”